# JQ-1 ameliorates schistosomiasis liver granuloma in mice by suppressing male and female reproductive systems and egg development of *Schistosoma japonicum*

**Jiaming Tian**[1,2], **Bingxin Dai**[1], **Li Gong**[1], **Pingping Wang**[1], **Han Ding**[1], **Siwei Xia**[3], **Weice Sun**[3], **Cuiping Ren**[1], **Jijia Shen**[1]\*, **Miao Liu**[1]\*

**1** Department of Microbiology and Parasitology, Anhui Provincial Laboratory of Microbiology and Parasitology, Anhui Key Laboratory of Zoonoses, School of Basic Medical Sciences, Anhui Medical University Hefei, Anhui, People's Republic of China, **2** Microbiological Laboratory, Anhui Provincial Center for Disease Control and Prevention, Hefei, Anhui, People's Republic of China, **3** The Second Clinical Medical College, Anhui Medical University, Hefei, Anhui, People's Republic of China

\* shenjijia@hotmail.com (JS); iammiaoliu@126.com (ML)

**Data Availability Statement:** All relevant data are within the manuscript and its Supporting Information files.

## Abstract

Schistosomiasis is a serious and widespread parasitic disease caused by infection with *Schistosoma*. Because the parasite's eggs are primarily responsible for schistosomiasis dissemination and pathogenesis, inhibiting egg production is a potential approach to control the spread and severity of the disease. The bromodomain and extra-terminal (BET) proteins represent promising targets for the development of epigenetic drugs against *Schistosoma*. JQ-1 is a selective inhibitor of the BET protein family. In the present study, JQ-1 was applied to *S. japonicum* in vitro. By using laser confocal scanning microscopy and EdU incorporation assays, we showed that application of JQ-1 to worms in vitro affected egg laying and the development of both the male and female reproductive systems. JQ-1 also inhibited the expression of the reproductive-related genes *SjPlk1* and *SjNanos1* in *S. japonicum*. Mice infected with *S. japonicum* were treated with JQ-1 during egg granuloma formation. JQ-1 treatment significantly reduced the size of the liver granulomas and levels of serum alanine aminotransferase and aspartate aminotransferase in mice and suppressed both egg laying and the development of male and female *S. japonicum* reproductive systems in vivo. Moreover, the mRNA expression levels of some proinflammatory cytokines were decreased in the parasites. Our findings suggest that JQ-1 treatment attenuates *S. japonicum* egg–induced hepatic granuloma due at least in part to suppressing the development of the reproductive system and egg production of *S. japonicum*. These findings further suggest that JQ-1 or other BET inhibitors warrant additional study as a new approach for the treatment or prevention of schistosomiasis.

**Funding:** This work was supported by grants from the National Natural Science Foundation of China (http://www.nsfc.gov.cn) (grant numbers 81271865) and Key University Science Research Project of Anhui Province of China (KJ2019A0223). The funders had no role in study design, data collection and analysis, decision to publish, or preparation of the manuscript.

**Competing interests:** The authors have declared that no competing interests exist.

## Author summary

Among neglected tropical diseases, schistosomiasis is a serious disease caused by infection with the parasite *Schistosoma japonicum*. Treatment of schistosomiasis is currently almost exclusively with praziquantel, which kills mainly adult parasites, with minimal effectiveness against immature schistosomes and eggs. However, the parasite's eggs are primarily responsible for schistosomiasis dissemination and pathology. In addition, overuse of praziquantel in epidemic areas has led to drug resistance and a reduced cure rate. Thus, new parasite targets for the development of novel therapeutics are crucial. Here, we evaluated the potential of JQ-1, a bromodomain and extra-terminal protein inhibitor, to suppress the production of *S. japonicum* eggs. Application of JQ-1 to *S. japonicum* in vitro decreased the number of mature germ cells, the rates of oviposition, and the number of eggs produced in each male-female pairing. JQ-1 treatment of mice infected with *S. japonicum* ameliorated hepatic granuloma and decreased serum liver enzymes, suggesting improved liver function. These results indicate that JQ-1 inhibits reproductive development and egg production in *S. japonicum*, providing supporting evidence that JQ-1 warrants additional study for use as a novel approach in the prevention or treatment of schistosomiasis.

## Introduction

Schistosomiasis is an acute and chronic parasitic disease caused by infection with *Schistosoma*, a parasite that is endemic in 78 countries and is responsible for approximately 280,000 deaths each year [1]. In China, zoonotic schistosomiasis caused by *S. japonicum* is major public health threat affecting more than a million people and hundreds of thousands of livestock [2].

Praziquantel is a widely used, high-efficiency, broad-spectrum, oral antiparasitic drug for the treatment of various forms of schistosomiasis, but praziquantel kills only adult worms and is minimally effective against immature schistosomes and eggs [2–3] In addition, the repeated and large-scale use of praziquantel in epidemic areas has led to drug resistance and a reduced cure rate [4–5]. Thus, there is an urgent need to identify new targets for the development of novel parasitic therapeutics. Owing to the key role of fertilized eggs in maintaining the life cycle and inducing pathogenesis [2–3], blocking egg production is a potential alternative approach to control the occurrence, development, and spread of schistosomiasis.

The bromodomain and extra-terminal (BET) family of proteins specifically recognizes acetylated lysine residue sites and participates in the regulation of epigenetic protein expression, which plays a key role in regulating various biological processes [6]. JQ-1 is a selective inhibitor of BET family proteins and has been shown to have promising anti-tumor and anti-inflammatory effects [7]. In a pilot study, we used JQ-1 to treat hepatic granuloma caused by infection with *S. japonicum*. Mice infected with S. *japonicum* cercariae were injected intraperitoneally with JQ-1 (50 mg/kg) during egg granuloma formation. Unexpectedly, JQ-1 significantly reduced the sizes of the liver granuloma and egg burden; however, JQ-1 treatment had no effect on worm load. We hypothesized that JQ-1 would be effective in inhibiting egg production in *S. japonicum* and sought to explore the mechanisms underlying this effect.

Thus, the aim of the present study was to confirm that JQ-1 reduces egg production of *S. japonicum* and to investigate the potential mechanisms undergirding this effect. To that end, we applied JQ-1 to schistosomes in vitro and assessed the effects on their reproductive development and egg production. We also treated C57BL/6 mice infected with *S. japonicum* with JQ-1 to assess the effects of the drug on hepatic granuloma and liver function. Our findings

indicated that JQ-1 inhibited the reproductive development of male and female *S. japonicum* and reduced egg production of the parasite while ameliorating hepatic granuloma and improving liver function in infected mice. These findings lay a foundation for further study to develop JQ-1 or other BET inhibitors as a new approach for the treatment and prevention of schistosomiasis.

## Materials and methods

### Ethics statement

All experiments carried out on animals were conducted in accordance with and were approved by the Animal Ethics Committee of Anhui Medical University (approval No. LLSC20170247) and conformed to the guidelines outlined in the Guide for the Care and Use of Laboratory Animals. All infection was performed under anesthesia, and all efforts were made to minimize suffering.

### Animals and parasites

Female Kunming mice (6–8 weeks old) and female C57BL/6 mice (6–8 weeks old) were provided by the Experimental Animal Center of Anhui Province in Hefei, China. The mice were housed under specific pathogen-free conditions at Anhui Medical University. *Oncomelania hupensis* snails infected with *S. japonicum* (a Chinese mainland strain) were purchased from the Jiangxi Provincial Institute of Parasitic Diseases in China.

### Treatment of schistosomes with JQ-1 in vitro

Cercariae were shed in a beaker after exposing 30 *O. hupensis* infected with *S. japonicum* to sunlight for 4 h (25–28˚C). For mixed infections, cercariae released from several infected *O. hupensis* were used. Kunming mice were infected percutaneously with 80–90 cercariae and were humanely killed on the 28th day after infection. All paired parasites were harvested by perfusion and washed three times with RPMI-1640 medium. The worms were then cultured in vitro with RPMI-1640 (Gibco, Grand Island, NY, USA) at 37˚C and 5% $CO_2$. The RPMI-1640 medium was supplemented with 10,000 U/mL penicillin, 10 mg/mL streptomycin, 250 μg/mL amphotericin B (Sangon Biotech, Shanghai, China), 15% fetal calf serum (Gibco), and glutamine (Gibco). For each experiment, 15 pairs of *S. japonicum* were maintained in a 6-well plate (i.e., 15 pairs/well). JQ-1 (Cat. No. HY-13030, MedChem Express; USA), was dissolved in dimethyl sulfoxide (DMSO). In each experimental group, 15 paired parasites were incubated in 3 mL of medium and treated with different concentrations of JQ-1 (0 μM, 5 μM, 10 μM, and 15 μM). All parasites were cultured at 37˚C for 10 d, and culture media was changed every 24 h. During this time, the viability and morphology of parasites, worm pairings, and the number of eggs were observed and recorded.

### Confocal laser scanning microscopy (CLSM)

For morphological analysis, collected worms were fixed in a solution of alcohol (95%), formalin (3%), and glacial acetic acid (2%) for at least 24 h. Worms were stained in hydrochloric acid–carmine dye (Ourchem, Shanghai, China) for 17 h and then destained in acidic 70% ethanol until the worms turned light pink. The worms were dehydrated in a graded ethanol series (70%, 90%, and 100%), cleared in 50% xylene diluted in ethanol and in 100% xylene for 1 min each, mounted onto slides with neutral gum, sealed with cover glass, and laid flat to dry. The morphology of their reproductive organs was observed with a CLSM (ZEISS LSM 880,

Germany) using an emission wavelength of 488 nm. Images were captured and stored at 1024 × 1024 pixels.

### 5-ethynyl-2′-deoxyuridine (EdU)-incorporation assay

For EdU labelling and detection of proliferating cells, paired worms treated with JQ-1 and control worms were incubated with 10 mM of EdU in medium for 24 h. BeyoClick EdU-594 Cell Proliferation Kits (Beyotime, Shanghai, China) were used to detect EdU incorporation. Couples were separated, fixed, and stained as described above, with minor alterations. The couples were rinsed twice in PBS and stained with Hoechst 33342 (diluted 1:1000 in PBS) in the dark for 10 min at room temperature. The worms were examined by CLSM using a ZEISS LSM 880 confocal microscope at a wavelength of 405 nm (for Hoechst) and 543 nm (for Azide 594).

### Treatment of schistosomes with JQ-1 *in vivo*

Four weeks after mice were infected with *S. japonicum*, mice in the experimental group were injected intraperitoneally with JQ-1 (50 mg/kg body weight per day), and mice in the control group were injected intraperitoneally with vehicle, namely, (2-hydroxypropyl)-β-cyclodextrin (HP-β-CD; Cat. No. 778966, Sigma; USA) 10% (wt/vol), once daily for 15 d. Animals were humanely killed 24 h after the last administration. The parasites, serum, and liver from each mouse were collected for subsequent experimental analyses.

### Quantitative PCR

Total RNA from adult *S. japonicum* worms or the liver of each mouse was isolated using TRIzol Reagent (Life Technologies, Carlsbad, CA, USA). The total RNA concentration and purity were detected using a NanoDrop 2000 system (Thermo Fisher Scientific, USA). Total RNA (500 ng) from the worms was reverse transcribed to cDNA by using a PrimeScript RT Reagent Kit (TaKaRa, Dalian, China) according to the manufacturer's instructions. A reliable reference gene for transcriptome analysis of *S. japonicum*, *PSMD4*, was used as a control gene in the assays [8], and GAPDH was used as a control gene for transcriptome analysis of the liver. The experiment was carried out using the StepOnePlus Real-Time PCR System (Applied Biosystems, Foster City, CA, USA). The relative expression level of each gene was analyzed using SDS v.1.4 software (Applied Biosystems). The procedure for quantitative PCR analysis was conducted as described previously [9], and the primers were designed and synthesized by Sangon Biotech Co. Ltd. The PCR primer sequences are described in the S1 Table.

### Serum liver enzyme quantification

For assessment of mouse liver function, a serum aminotransferase test kit (Nanjing Jiancheng Bioengineering Institute, Nanjing, China) was used to measure the levels of serum alanine aminotransferase (ALT) and aspartate aminotransferase (AST), according to the manufacturer's instructions. The levels of serum ALT and AST are reported in units per liter.

### Egg count in liver tissue

Approximately 0.1 g of liver tissue was collected from the same area of the liver in each mouse and weighed. Potassium hydroxide (10%; 1 mL) was added to the liver tissue for digestion at 37°C for 2 h. The number of eggs in each sample was then counted using a light microscope.

### Histology and immunohistochemistry of liver sections

Fresh liver tissue (1.0 g) was fixed in 1% buffered formalin and embedded in paraffin. The deparaffinized tissue sections were affixed to slides, and sections (thickness, 4 μm) were stained with hematoxylin and eosin and examined for quantitative and qualitative changes. Computer-assisted morphometric software (Image-Pro Plus; Media Cybernetics) was used to determine the total areas of the tissue and granuloma on each slide so that the area of the granulomas could be reported as a percentage of the total area for each slide. For each specimen, at least three non-continuous slides were measured, and the mean values obtained from eight mice in each group were used for statistical analysis.

### Statistical analysis

Statistical analysis was performed using GraphPad Prism software (version 6.0). All data were obtained from three independent experiments, each using triplicate samples and following the same protocol. The statistical significance of the difference between two data sets was analyzed using Student's t-test, and one-way analysis of variance (ANOVA) was used for multiple comparisons, followed by Tukey's post hoc tests when appropriate. Data are presented as means ± SEM and were considered statistically significant for $P$-values $<0.05$.

## Results

### Effects of JQ-1 treatment on pairing rate and egg production

The number of male-female paired worms was counted on the 10th day of culture to determine the effect of JQ-1 treatment on the pairing rate. We found that the number of paired worms in the cultures treated with JQ-1 was similar to that in the control group treated with vehicle (Fig 1A). No significant changes in schistosome activity or in the number of viable worms were detected between the JQ-1-treated group and the control group. However, the number of eggs collected in the medium and counted using light microscopy was decreased in the cultures treated with JQ-1 compared with controls (Fig 1C–1F). To further analyze the effects of JQ-1 on egg production in the paired females, we counted egg numbers and found that compared with the DMSO-treated group, the number of eggs ($P < 0.001$) in the JQ-1-treated group decreased in a concentration-dependent manner (Fig 1B).

### JQ-1 treatment decreases mitotic activity in somatic and germ cells

We investigated whether JQ-1 affects mitosis in *S. japonicum* by performing EdU-incorporation assays using JQ-1-treated worms to assess cell proliferation. Worm pairs treated with JQ-1 for 10 d exhibited a substantial decrease in the number of EdU-labeled cells in the gonads, parenchyma, and subtegument of both sexes. In the untreated control group, a substantial number of EdU-labeled cells were detected in the vitellarium and ovary of adult females as well as in the testis and parenchyma of adult males (Fig 2), which indicated high mitotic activity in these organs. Adult worms treated with JQ-1 for 10 d showed a slight decrease in the number of EdU-positive cells in the vitellarium of females and the testis and parenchyma of males; greater decreases were observed with increasing concentrations of JQ-1. At the highest concentration, JQ-1–treated worm organs and tissues had almost no EdU-labeled cells (Fig 2D and 2H).

### Effects of JQ-1 treatment on reproductive organ development

Consistent with the observed decreased egg production, CLSM analyses of worm pairs treated with JQ-1 revealed morphologic abnormalities in the gonads of both sexes. After treatment for

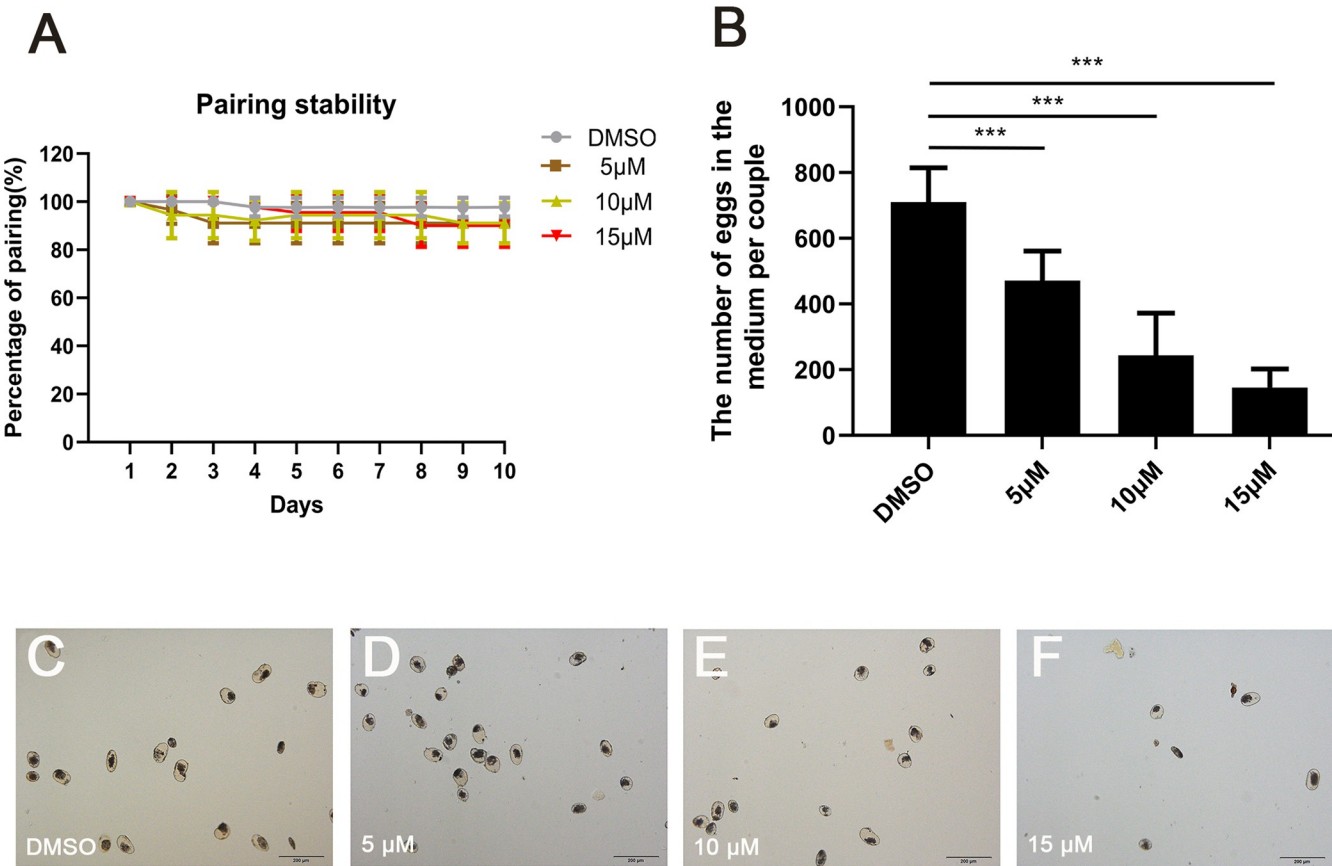

**Fig 1. Effect of JQ-1 on male-female pairing rate, egg production, and egg morphology in *S. japonicum*.** Effects of different concentrations (5 μM, 10 μM, and 15 μM) of JQ-1 application on male-female pairing stability (A), egg production (B), and egg morphology (C-F) in *S. japonicum* pairs cultured in vitro for 10 d. Data represent the mean ± SEM of three independent experiments. Scale bars: 200 μm. Asterisks show statistical differences (***$P < 0.001$) tested by one-way ANOVA with multiple comparisons (Tukey's post-hoc test).

10 d, no morphological anomalies were observed in the testes and vesicles of the males (Fig 3A and 3E) or the ovaries of the females (Fig 4A) in the control group. The testes of DMSO-treated male schistosomes were composed of several testicular lobes arranged bead-like, and each testicular lobe contained a large number of spermatocytes and spermatogonia at different stages (Fig 3A–3D), The size of the testicular lobes in the group treated with the high concentration of JQ-1 was much smaller than that in the DMSO-treated group, and the numbers of spermatogonia and spermatocytes in the male testes were significantly reduced and more loosely arranged (Fig 3D). The results of CLSM (Fig 3E–3H) showed that the number of spermatozoa in the seminal vesicles of schistosomes in the JQ-1-treated group was reduced, and the development of the spermatozoa was impaired. Compared with controls, the group treated with JQ-1 showed a markedly reduced diameter of the testicular lobes (Fig 3I), which was paralleled by a reduction in cell density within the testes as well as by empty seminal vesicles. In the group treated with JQ-1, the morphology of whole germ cells in both the testis and ovary were markedly changed. Those changes were more obvious with increasing concentrations of JQ-1. The ovaries of the DMSO-treated female schistosomes were composed of small immature oocytes in the anterior part and larger primary oocytes in the posterior part, the sizes of the primary oocytes and immature oocytes were reduced, and the cells of the JQ-1-treated groups were not as full as the cells of DMSO-treated groups (Fig 4A–4D). The length and

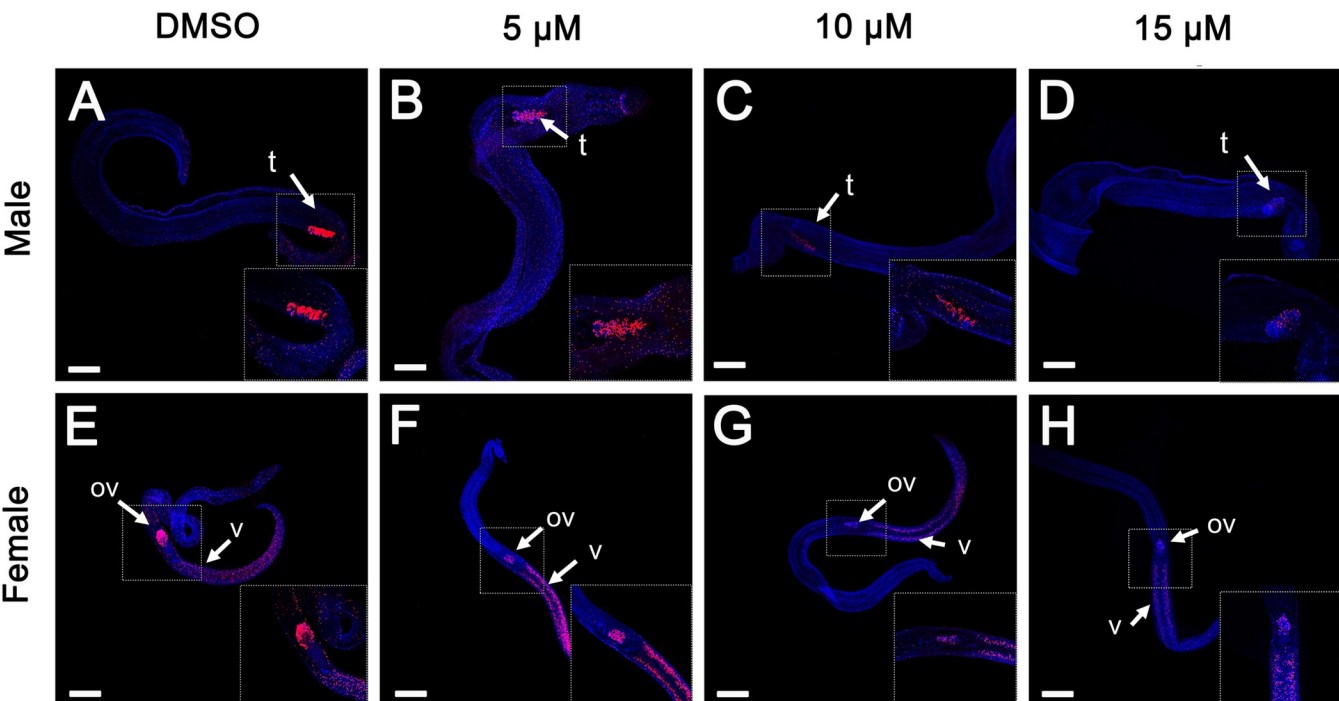

**Fig 2. Effect of JQ-1 on cell proliferation in male-female pairs of *S. japonicum*.** Red signals indicate active mitotic cells labeled by EdU; blue signals, Hoechst-positive cells. (A-D) Male *S. japonicum* and (E-H) female *S. japonicum*. EdU-incorporated cells are detected in the testes and parenchyma of untreated males (A) and in the vitellarium and ovary of untreated females (E). EdU-positive cells are detected after application of JQ-1 at 5 μM (B, F), 10 μM (C, G), and 15 μM (D, H). Abbreviations: ov, ovary; t, testes; SV, seminal vesicles. Scale bars: 200 μm.

width of the ovaries in females treated with JQ-1 were significantly smaller than those of untreated controls (Fig 4I–4K). Large pore-like structures were observed in the testes and ovaries of males and females, respectively (Figs 3 and 4). These morphological changes in both females and males were greatest after treatment with the highest concentration (15 μM) of JQ-1.

## JQ-1 treatment decreases *SjNanos1*, *SjPlk1* mRNA levels

To explore the mechanisms undergirding the observed effects of JQ-1 on *S. japonicum*, we used quantitative PCR to detect the levels of the *S. japonicum* protein coding genes polo-like kinase 1 (*SjPlk1*) and *SjNanos1*, two genes related to schistosome reproduction, after application of different concentrations of JQ-1 in vitro. Compared with the control group, the expression levels of *SjNanos1* mRNA in the JQ-1-treated worms were down-regulated in both males (Fig 5A) and females (Fig 5B), and this effect was more marked with increasing concentrations of JQ-1. Similarly, the expression levels of *SjPlk1* mRNA were also down-regulated in both males and females, and this effect was also more marked with increasing concentrations of JQ-1 (Fig 5C and 5D). However, JQ-1 treatment did not affect the expression of *SjBrd2* mRNA (Fig 5E and 5F).

## JQ-1 ameliorates liver granuloma caused by *S. japonicum* infection

In the fourth week after *S. japonicum* infection, mice in the experimental group were injected with JQ-1, and mice in the control group were injected with the vehicle HP-β-CD, once daily for 15 d. All mice were humanely killed after 15 d of treatment (Fig 6A). As shown in Fig 6B,

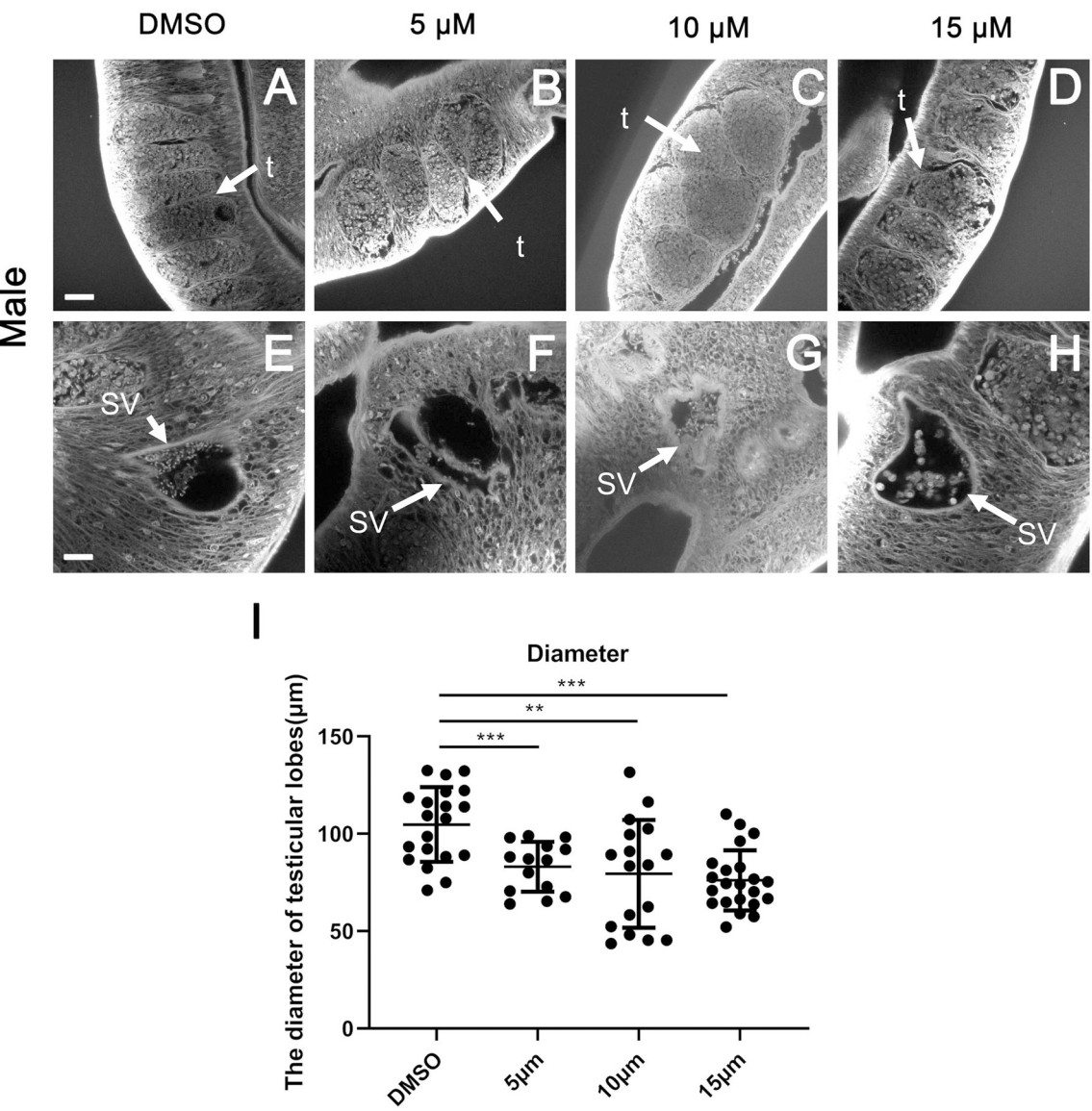

**Fig 3. Morphological changes of spermatozoa in testes and seminal vesicles of *S. japonicum* treated with JQ-1 in vitro.** Worms were stained with carmine hydrochloride and analyzed using confocal laser scanning microscopy. (A, E) Control worms; worms treated with JQ-1 at 5 μM (B, F), 10 μM (C, G), and 15 μM (D, H). (A-D) Scale bars: 20 μm; (E-H) Scale bars: 10 μm. Abbreviations: t, testis; SV, sperm vesicle. (I) Comparison of the diameter of the testicular lobes after JQ-1 application at the indicated concentration for 10 d. Data represent the mean ± SEM (n ≥ 15 for each group). Asterisks show statistical differences (**$P < 0.01$; ***$P < 0.001$) tested by one-way ANOVA with multiple comparisons.

livers obtained from mice in the HP-β-CD group had large agglomeration, and granuloma inflammation was severe. However, there was marked reduction of liver surface granulomatous nodules in the JQ-1-treated group. The livers obtained from mice in the JQ-1-treated group were lighter and more vivid in color, and the surface was relatively smooth, compared with the livers from mice in the control group. Hematoxylin and eosin staining of the liver showed that the percentage of the area of the liver that had granulomas in the JQ-1-treated group was significantly reduced compared with that in the HP-β-CD control group (Fig 6C) ($P < 0.01$). In addition, the weights of the liver and spleen obtained from mice treated with JQ-1 were significantly lower than those from control mice (Fig 6D and 6E). Moreover, the

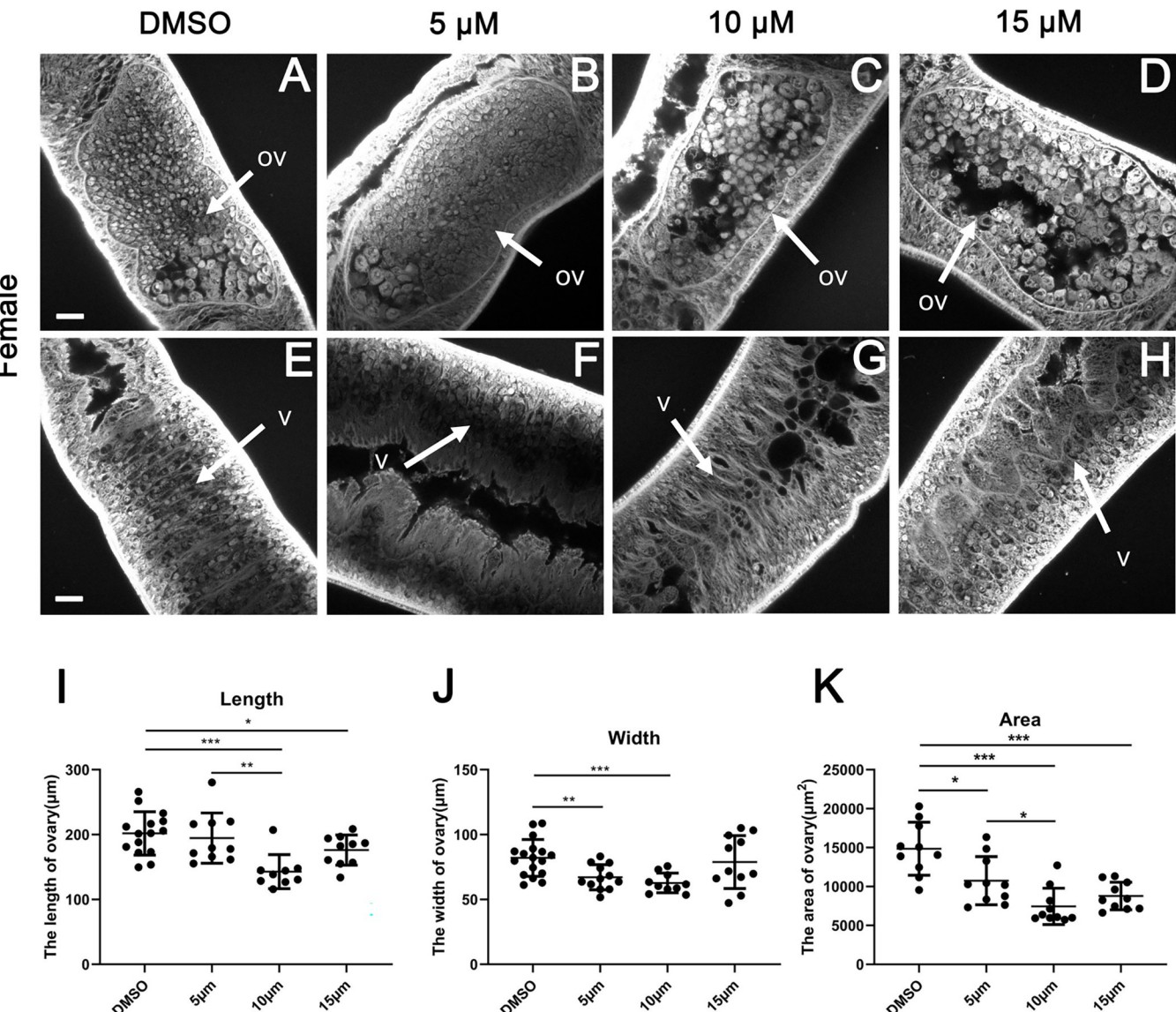

**Fig 4. Morphological changes of ovaries and yolk glands in female *S. japonicum* treated with JQ-1.** Worms were stained with carmine hydrochloride and analyzed using confocal laser scanning microscopy. (A, E) Control worms; worms treated with JQ-1 at 5 μM (B, F), 10 μM (C, G), and 15 μM (D, H). Abbreviations: ov, ovary; v, vitellarium; (A-H) Scale bars: 20 μm. Comparison of the length, width, and area of the ovary after JQ-1 application at the indicated concentration (I-K) for 10 d. Data represent the mean ± SEM (n ≥ 15 for each group). Asterisks show statistical differences (**$P < 0.05$, **$P < 0.01$, ***$P < 0.001$) tested by one-way ANOVA with multiple comparisons.

results of the AST and ALT assays showed that the activity of serum transaminase in the JQ-1–treated group was significantly lower than that in the control group (Fig 6F) ($P < 0.05$). To further explore the effect of JQ-1 treatment to ameliorate hepatic granuloma in mice infected with *S. japonicum*, we used quantitative PCR to detect the expression levels of a series of inflammatory factors. The mRNA expression levels of the genes in the HP-β-CD–treated control group were set at 1. As shown in Fig 6G, the mRNA expression levels of the inflammatory factors in the JQ-1-treated group relative to those in the control group were significantly decreased ($P < 0.05$). Notably, the expression level of interleukin 13, an inflammatory factor closely related to the formation of granuloma caused by *S. japonicum*, was significantly reduced.

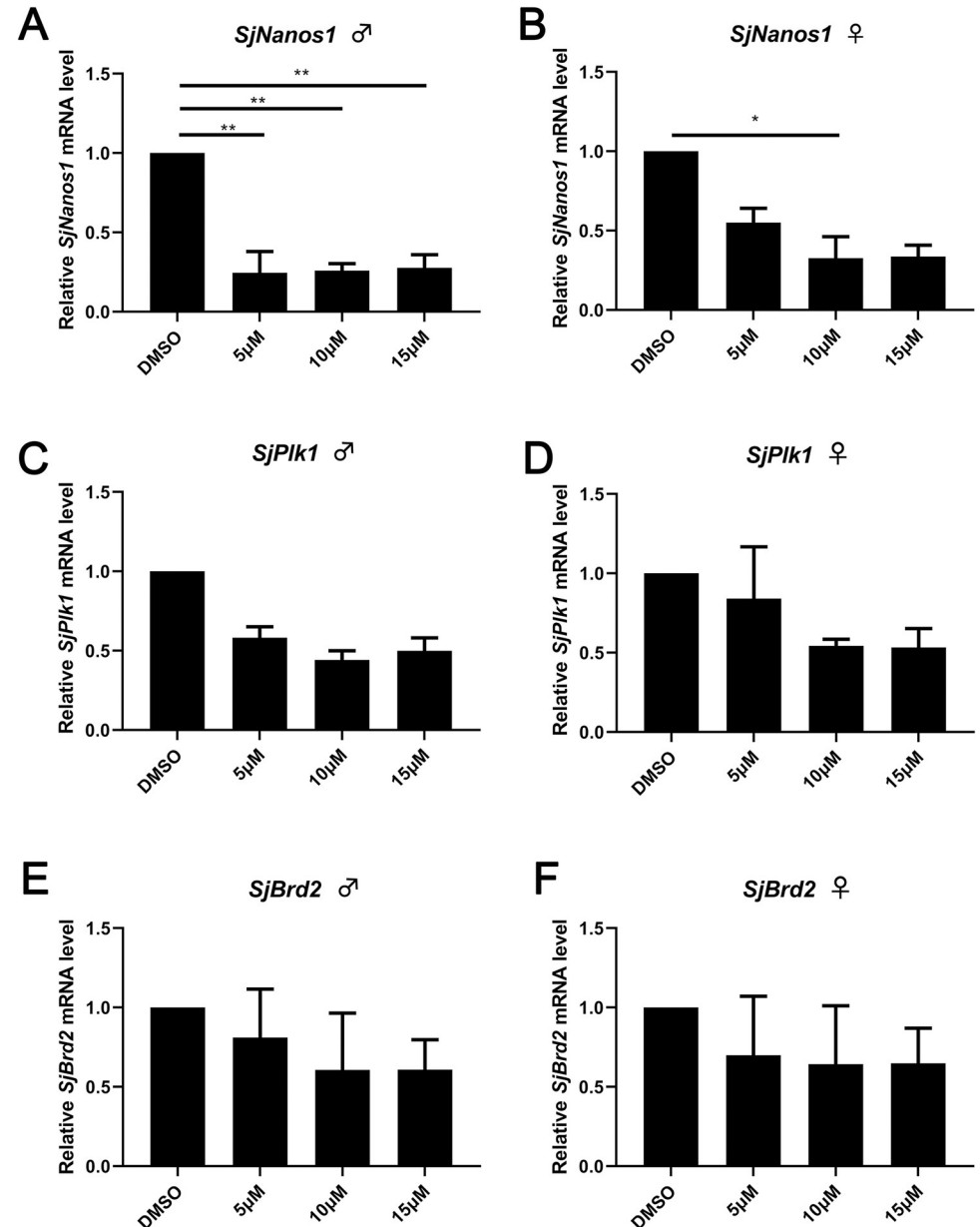

**Fig 5. Results of quantitative PCR analyses of *S. japonicum* cultured with or without JQ-1 for 10 d.** Relative transcription level of *Nanos1* in male (A) and female *S. japonicum* (B). Relative transcription level of *Plk1* in male (C) and female (D) *S. japonicum*. Relative transcription level of *Brd2* in male (E)€ and female (F) *S. japonicum*. Data represent the mean ± SEM of three independent experiments. Asterisks show statistical differences (*$P < 0.05$; **$P < 0.01$) tested by one-way ANOVA with multiple comparisons.

## Effects of JQ-1 treatment on schistosome eggs in the liver and on adult worms in mice infected with *S. japonicum*

The above results suggested that JQ-1 alleviated liver injury caused by schistosome infection to some extent and reduced the formation of hepatic granuloma in mice. To observe whether JQ-1 affected *S. japonicum* eggs in the liver, we evaluated the quantity of eggs in the liver of mice in the JQ-1-treated group compared with that in the HP-β-CD-treated control group after

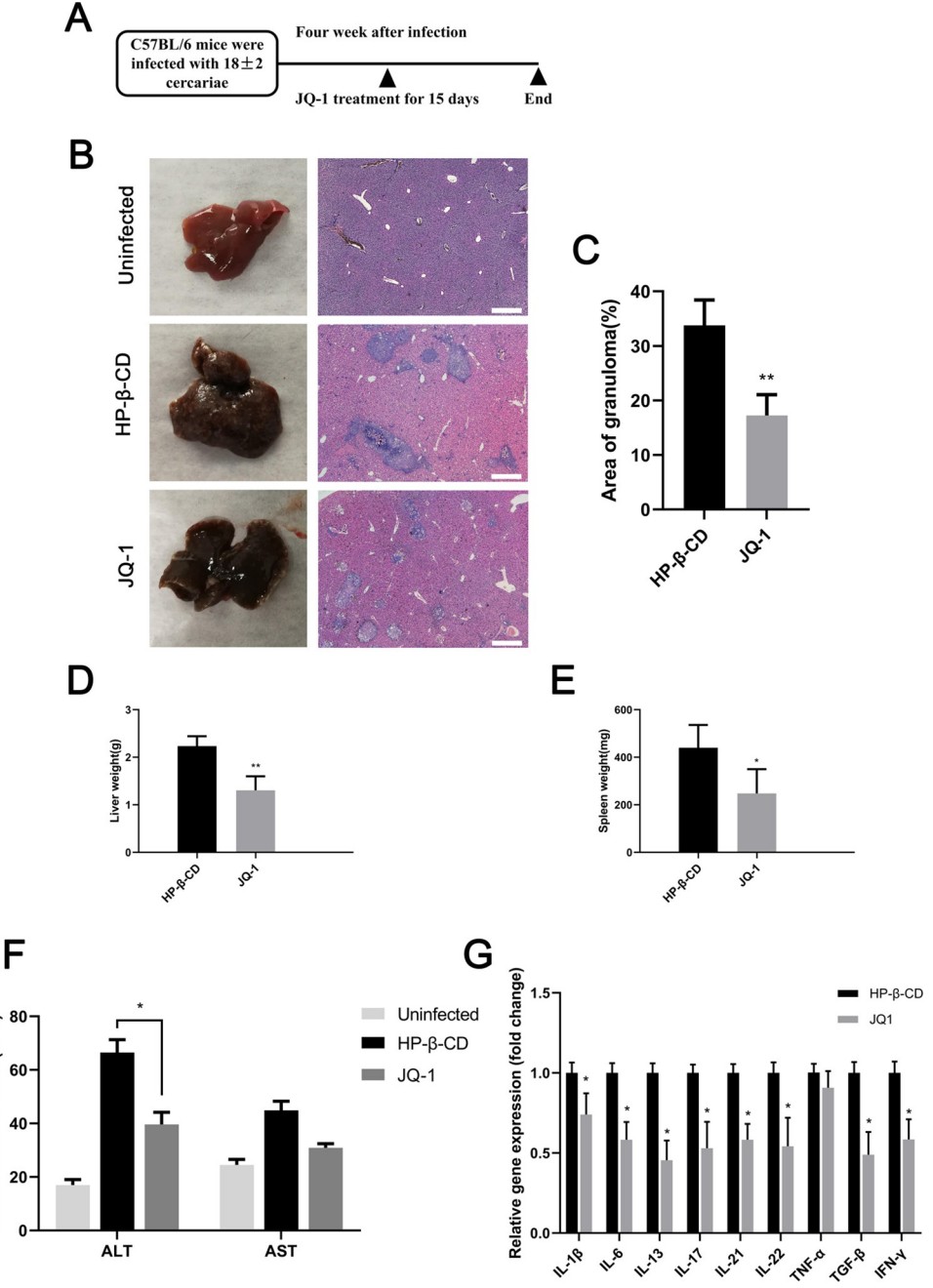

**Fig 6. Effect of JQ-1 treatment on liver granuloma in mice infected with *S. japonicum*.** (A) Protocol used to assess liver granuloma in mice. (B) Gross appearance of livers obtained from mice infected with *S. japonicum* and treated with JQ-1 or vehicle (HP-β-CD). Liver slices stained with hematoxylin and eosin. Scale bars: 500 μm. (C) Measurement of granuloma area as a percentage of total area as assessed by computer-aided morphometry. (D) Liver weights of *S. japonicum* infected mice treated with JQ-1 or HP-β-CD. (E) Spleen weights of *S. japonicum* infected mice treated with JQ-1 or HP-β-CD. (F) Effect of JQ-1 treatment on serum alanine aminotransferase (ALT) and aspartate aminotransferase (AST) in mice infected with *S. japonicum*. (G) Effect of JQ-1 treatment on mRNA expression of inflammation-related genes in the liver of mice infected with *S. japonicum*. Data represent the mean ± SEM (n = 9 for each group). Asterisks denote statistically significant differences (Student's t-test, *$P < 0.05$; **$P < 0.01$) vs. the HP-β-CD–treated control group.

schistosome infection. The liver tissue obtained following digestion with 10% potassium hydroxide was used to observe the morphology of the eggs and to count them. We found that the proportion of abnormally small or dead eggs was increased in the JQ-1-treated group (Fig 7A and 7B). The volume of eggs in the liver of JQ-1-treated infected mice was approximately 40% lower than that of control mice injected with HP-β-CD (Fig 7B). By contrast, the numbers

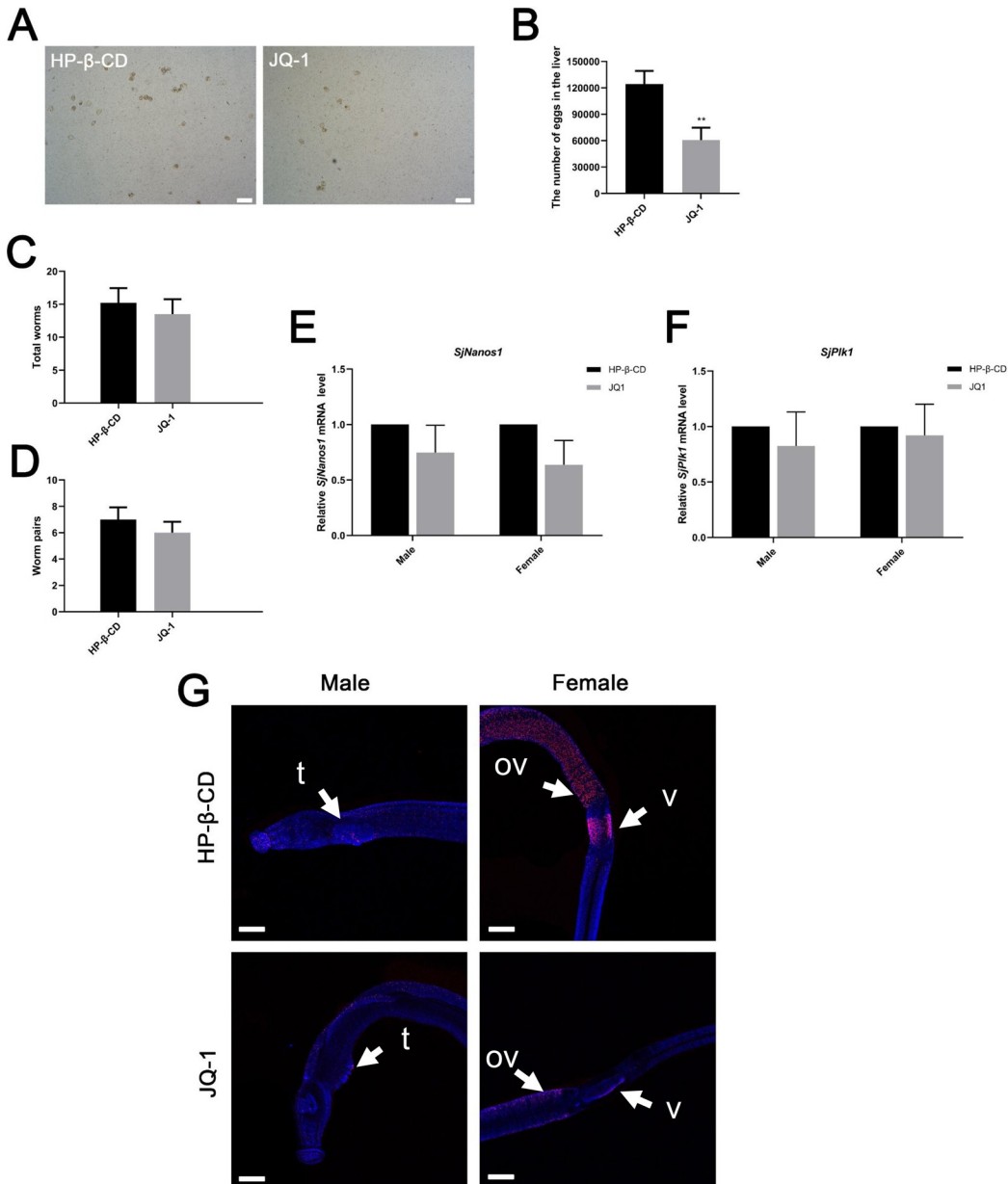

**Fig 7. JQ-1 treatment alters germ cell proliferation of *S. japonicum* and egg production in the liver of mice infected with *S. japonicum*.** (A) Egg morphology and (B) production in the liver. (C) Numbers of adult worms and (D) worm pairs in the liver. **(E-F) Results of quantitative PCR analyses of *S. japonicum* detected in *S. japonicum*–infected mice treated with JQ-1. (G)** Red signals indicate active mitotic cells labeled by EdU; blue, Hoechst-positive cells. EdU-incorporated cells in control worms were detected in the testes and parenchyma of males and in the vitellarium and ovary of females. (A) Scale bars: 500 μm. (G) Abbreviations: ov, ovary; t, testes. Scale bars: 100 μm. Data represent the mean ± SEM (n = 9 for each group). Asterisks denote statistically significant differences (Student's t test, ** $P < 0.01$).

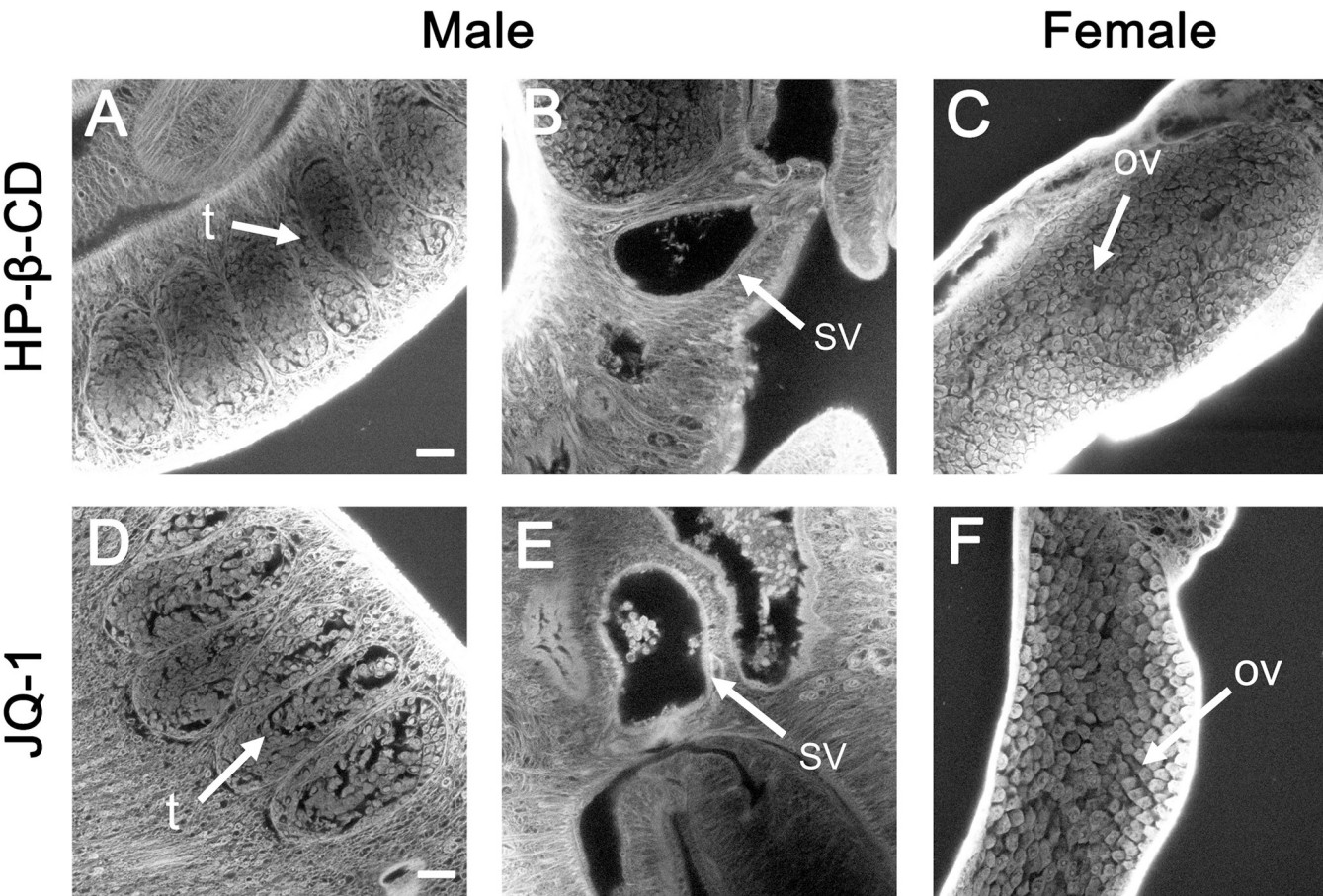

**Fig 8. Morphological changes in the testis and ovary of *S. japonicum* treated with JQ-1 in vivo.** Worms were stained with carmine hydrochloride and analyzed using confocal laser scanning microscopy. (A-C) Testes, seminal vesicles, and ovary of worms in control mice. (D-F) Testes, seminal vesicles and ovary of worms in mice treated with JQ-1. Abbreviations: ov, ovary; t, testes; SV, seminal vesicles. Scale bars: 20 μm.

of adult worms and worm pairs in the livers of the JQ-1–treated group were not affected (Fig 7C and 7D). Although this effect in the treated group may have been due to the significant decrease in the number of eggs or to the increase in the number of small or dead eggs, it may also be related to the immune regulation of JQ-1 in mice. JQ-1 treatment decreased the expression of *SjNanos1*and *SjPlk1* mRNA, but there was no statistically significant difference between the treated group and the control group (Fig 7E and 7F). We used an EdU-incorporation assay to assess the proliferation of germ cells in schistosomes of infected mice (Fig 7G). Although some differences between the control group and the treated group were observed, the differences were not as obvious as those observed in the in vitro experiments.

CLSM analyses of the JQ-1-treated group revealed morphologic abnormalities in the gonads of both sexes. In the control HP-β-CD-treated group, no morphological anomalies were observed in the testes of the males (Fig 8A) or the ovaries of the females (Fig 8C). By contrast, compared with the control group, the number of spermatozoa in the seminal vesicles of schistosomes in the JQ-1-treated group was reduced and the development of spermatozoa was impaired (Fig 8B and 8E). In addition, in JQ-1 treatment group, the overall morphology of the germ cells of schistosomes in both the testis and ovary were markedly changed. Moreover, large pore-like structures could be found in the testes and ovaries of male and female schistosomes, respectively, in the JQ-1-treated group (Fig 8D and 8F).

## Discussion

The present study assessed the effects of JQ-1 application on *S. japonicum* in vitro and in vivo and investigated the potential mechanisms undergirding the observed effects. The results of our in vitro studies indicated that although JQ-1 application did not affect the number or pairing of adult schistosomes, the number of eggs decreased in a concentration-dependent manner. In addition, mitotic activity in the somatic and germ cells of the adult worms decreased. The numbers of spermatogonia and spermatocytes were significantly decreased and the testicular lobes were significantly smaller in male schistosomes treated with JQ-1 compared with schistosomes in the control group. Moreover, large pore-like structures were observed in the testes and ovaries of JQ-1-treated schistosomes. These results suggested that JQ-1 specifically inhibited the proliferation of germ cells. Our EdU incorporation assays confirmed that JQ-1 reduced the number of proliferating cells in both the ovaries and testes of schistosomes. Proliferation of those cells is essential for the initiation and continuous production of mature germ cells. Treatment with JQ-1 also decreased the expression levels of two genes related to schistosome reproduction, *SjPlk1* and *SjNanos1*, in a concentration-dependent manner. Thus, this study is the first, to our knowledge, to show that JQ-1 is effective against reproductive development and egg production of adult *S. japonicum* in vitro. In schistosomiasis in humans, morbidity is mainly attributed to the eggs because of the granulomatous inflammatory reaction caused by the host immune response to egg antigens [2–3]. Thus, we assessed the ability of JQ-1 to treat hepatic granuloma in mice infected with *S. japonicum* in vivo. JQ-1 treatment significantly decreased the percentage of the area of the liver with granulomas, the activity of liver serum transaminase, and schistosome egg production in the liver of mice without affecting the survival of adult worms. The attenuated egg production was accompanied by decreased expression levels of proinflammatory cytokines, which may have contributed to the amelioration of hepatic granuloma. Taken together, our findings provide evidence supporting the development of JQ-1 as an anti-schistosomal agent.

The BET family proteins are characterized by the presence of two tandem bromodomains and an extra-terminal domain, which are found in BRD2, BRD3, BRD4, and BRDT in mammals [6]. The domain organization of mammalian BET proteins is conserved in orthologs, including in Drosophila FSH and *Saccharomyces cerevisiae* Bdf1 and Bdf2. Bromodomains that specifically bind acetylated lysine residues in histones serve as chromatin-targeting modules that decipher the histone acetylation code. BET proteins play a crucial role in regulating gene transcription through epigenetic interactions between bromodomains and acetylated histones during cell proliferation and differentiation [10–11]. *Brd2* mRNA is express in distinct patterns during ovarian folliculogenesis, which is essential for embryonic development in the mouse [12–13]. Brdt acetylated histone H4-dependent chromatin remodeling in mammalian spermiogenesis is essential for male germ cell differentiation [14–15]. In addition, a BRDT-like function in Drosophila plays crucial roles in spermatid differentiation [16]. Epigenetic modifications, including DNA methylation, histone modifications, and non-coding RNAs, play important roles in the development and reproduction of schistosomes [17]. SmGCN5 and SmCBP1 are two histone acetyltransferases in *Schistosoma mansio*n. The knockdown of SmGCN5 or SmCBP1 significantly inhibits Smp14 expression, which compromises the reproductive system of mature females, egg-laying, and egg morphology [18]. Sirtuins are a family of histone deacetylases, and sirtuin inhibitors can inhibit apoptosis and death in schistosome larvae, disruption of adult worm pairs, inhibition of egg laying, and damage to male and female worm reproductive systems [19–20].

As a first-in-class, potent, and selective inhibitor of the BET signaling pathway, JQ-1 has been widely used in biology studies. The results of some of those numerous studies indicate

that JQ-1 interacts with the BRD pocket in a manner competitive with acetylated peptide binding, resulting in the displacement of BET proteins from acetylated chromatin in cells exposed to these inhibitors and disruption of their associated transcript initiation and elongation factors. JQ-1 has also been used as a pharmacological tool for elucidating the roles and functions of BET in mammals. However, little is known about the effect of JQ-1 on parasites.

Nanos has been described as a necessary factor in the differentiation and migration of primordial germ cells, which play an essential role in the proliferation of germ cells in schistosomes [21–22]. SmPlk1 regulates the cell cycle G2/M transition in *Xenopus* oocytes, which is important for cell-cycle progression in the gonadal cells of *Schistosoma* [23–24]. In the present study, we investigated whether JQ-1 also affected the transcript level of *Nanos1* and *Plk1*. Indeed, treatment with JQ-1 significantly reduced the transcript levels of both these genes in male and female worms, which likely affected the proliferation of the gonadal cells in *Schistosoma*.

This study has limitations that should be considered when interpreting our results. On the basis of previous publications [25–26], we used only a single dose of JQ-1 (50 mg/kg) to treat mice infected with *S. japonicum* for 15 d. Thus, we were unable to make any comparisons of the effects after various treatment times or dosage on parasites in infected mice. Future studies are needed to find the optimum therapeutic dosage.

In conclusion, our data showed that JQ-1 treatment ameliorated *S. japonicum* egg–induced hepatic granuloma, which may be due in part to suppressing the development of both the male and female reproductive systems and female egg production in this parasite. Our findings provide theoretical and practical evidence supporting the development of JQ-1 as an anti-schistosomal agent.

## Supporting information

**S1 Table. Sequences of quantitative PCR primers in this study.**
(DOCX)

## Acknowledgments

We thank a native English speaker at Onboard Editing for modifying the language in the manuscript.

## Author Contributions

**Conceptualization:** Jiaming Tian, Han Ding, Jijia Shen, Miao Liu.

**Data curation:** Jiaming Tian, Bingxin Dai, Li Gong, Pingping Wang, Han Ding, Siwei Xia, Cuiping Ren.

**Formal analysis:** Jiaming Tian, Bingxin Dai, Weice Sun.

**Funding acquisition:** Jijia Shen, Miao Liu.

**Investigation:** Jijia Shen.

**Methodology:** Jiaming Tian, Bingxin Dai, Li Gong, Pingping Wang, Han Ding, Siwei Xia, Weice Sun, Cuiping Ren, Miao Liu.

**Project administration:** Jijia Shen, Miao Liu.

**Resources:** Siwei Xia, Weice Sun, Cuiping Ren.

**Supervision:** Jijia Shen, Miao Liu.

 

**Writing – original draft:** Jiaming Tian, Miao Liu.

**Writing – review & editing:** Jiaming Tian, Bingxin Dai, Li Gong, Pingping Wang, Jijia Shen, Miao Liu.

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
