## [Decision Letter · Decision Letter 0]

27 Apr 2022

Dear Dr. Miao Liu,

Thank you very much for submitting your manuscript "JQ-1 ameliorates schistosomiasis liver granuloma in mice by suppressing male and female reproductive systems and egg development of Schistosoma japonicum" for consideration at PLOS Neglected Tropical Diseases. As with all papers reviewed by the journal, your manuscript was reviewed by members of the editorial board and by several independent reviewers. In light of the reviews (below this email), we would like to invite the resubmission of a significantly-revised version that takes into account the reviewers' comments. 

We cannot make any decision about publication until we have seen the revised manuscript and your response to the reviewers' comments. Your revised manuscript is also likely to be sent to reviewers for further evaluation.

Sincerely,

Alessandra Morassutti, PhD

Associate Editor

Maria Elena Bottazzi

Deputy Editor

Reviewer's Responses to Questions

**Key Review Criteria Required for Acceptance?**

**Methods**

-Are the objectives of the study clearly articulated with a clear testable hypothesis stated?

-Is the study design appropriate to address the stated objectives?

-Is the population clearly described and appropriate for the hypothesis being tested?

-Is the sample size sufficient to ensure adequate power to address the hypothesis being tested?

-Were correct statistical analysis used to support conclusions?

-Are there concerns about ethical or regulatory requirements being met?

Reviewer #1: (No Response)

Reviewer #2: (No Response)

**Results**

-Does the analysis presented match the analysis plan?

-Are the results clearly and completely presented?

-Are the figures (Tables, Images) of sufficient quality for clarity?

Reviewer #1: (No Response)

Reviewer #2: (No Response)

**Conclusions**

-Are the conclusions supported by the data presented?

-Are the limitations of analysis clearly described?

-Do the authors discuss how these data can be helpful to advance our understanding of the topic under study?

-Is public health relevance addressed?

Reviewer #1: (No Response)

Reviewer #2: (No Response)

**Editorial and Data Presentation Modifications?**

Reviewer #1: (No Response)

Reviewer #2: (No Response)

**Summary and General Comments**

Reviewer #1: The author studied the effect of JQ-1 on reproductive systems and egg development of Schistosoma japonicum to ameliorates liver granuloma in mice. The authors detected the male-female pairing rate, egg production, reproductive system and the expression of related genes of Schistosoma japonicum after JQ-1 treatment in vivo and in vitro. The data suggested JQ-1 treatment ameliorated S. japonicum egg–induced hepatic granuloma and suppressing the development of both the male and female reproductive systems and female egg production. This manuscript be of interest. However, there are some concerns about this paper.

Major concerns

1. JQ-1 is the inhibitor of BET. The mRNA and protein expression levels of BET protein family should be detected in this study. Authors may provide such data in different groups with or without JQ-1 treatment. 

2. Please include some positive controls for JQ-1 treatment in in vivo and in vitro models. 

3. No negative control group (normal control) in In vitro experiment

4. Effect of JQ-1 on cell apoptosis in male-female pairs of S. japonicum should be included.

5. Why did the author treat the S.japonicum infected mice with JQ-1 after four weeks’ infection? Please provide the reasons in the manuscripts.

6. It would be nice to detect the levels of SjNanos1, SjPlk1 mRNA in vivo.

7. JQ-1 has also been used as a pharmacological tool for elucidating the roles and functions of BET in mammals. How to evaluate the effect of JQ-1 on host mice? Do JQ-1 has the toxic effects on mice, especially hepatotoxicity? And how about the expression of inflammatory factors in mice treated with JQ-1 alone. 

8. The method of egg count in liver tissue is not clear. Because eggs deposited in liver are not evenly in schistosomiasis mice, liver tissues should be taken in the fix sites in each mouse in control group and test group.

9. The Figures problems:1) Illustrations for Fig 3 and Fig 4 were misplaced; 2) Fig 4, the labeled texts on axes in graphs are blurred and indistinct. 3) Fig5: the present graphs are unreasonable; the text labeled on axes are not accurate. 4）Fig7: graphs should be combined. 5) the morphological photos in Fig2,3,4,8,9 should be marked with test groups in figures and provided more illustration details.

Minor errors:

10. The English of the manuscript needs to be carefully edited.

11. P1 line15-16: Schistosomiasis is a serious and widespread parasitic disease caused by infection with Schistosoma japonicum._

12. Revised：…….by infection with Schistosoma.

13. P2 line38-39: schistosomiasis is a serious disease caused by infection with the parasite Schistosomiasis japonicum.

14. Revised：schistosomiasis japonicum is a serious disease caused by infection with the parasite S. japonicum.

15. P3 line58: S. japonicum. — Use italics and check the whole text especially in reference part.

16. P4 line74: Schistosoma japonicum — revised: S. japonicum. and check the whole text.

Reviewer #2: In this study, the effects of JQ-1, a selective inhibitor of BET protein family, on adult worm development of Schistosoma japonicum, especially reproductive system development, were investigated. The results showed that JQ-1 could reduce the egg laying and the germ cell division in the adult worms in vitro and in vivo, which resulted in the decrease of egg granuloma formation and release of proinflammatory factors in liver of the infected mice. However, when considering the use of JQ-1 as an anti-schistosomiasis drug, the authors do not answer or ignore two important questions: 1.The fate of eggs treated with JQ-1, that is, whether they can still be excreted from the intestine. 2. Whether or not JQ-1 has the effects on host germ cells? Otherwise, it is not practical or reasonable to evaluate the pharmacological effect of JQ-1 on Schistosoma japonicum reproduction. Although the authors also observed that JQ-1 inhibited the expression of reproductive genes SjPlk1 and SjNanos1 in the worms, they did not discuss the relationship between these effects and the known influence of JQ-1 on the function of mammalian BET proteins.

PLOS authors have the option to publish the peer review history of their article (what does this mean?). If published, this will include your full peer review and any attached files.

Reviewer #1: No

Reviewer #2: No
---

## [Decision Letter · Decision Letter 1]

13 Jul 2022

Dear Miao Liu,

We are pleased to inform you that your manuscript 'JQ-1 ameliorates schistosomiasis liver granuloma in mice by suppressing male and female reproductive systems and egg development of Schistosoma japonicum' has been provisionally accepted for publication in PLOS Neglected Tropical Diseases.

Best regards,

Alessandra Morassutti, PhD

Academic Editor

Maria Elena Bottazzi

Section Editor

Reviewer's Responses to Questions

**Key Review Criteria Required for Acceptance?**

**Methods**

-Are the objectives of the study clearly articulated with a clear testable hypothesis stated?

-Is the study design appropriate to address the stated objectives?

-Is the population clearly described and appropriate for the hypothesis being tested?

-Is the sample size sufficient to ensure adequate power to address the hypothesis being tested?

-Were correct statistical analysis used to support conclusions?

-Are there concerns about ethical or regulatory requirements being met?

Reviewer #1: (No Response)

Reviewer #2: (No Response)

**Results**

-Does the analysis presented match the analysis plan?

-Are the results clearly and completely presented?

-Are the figures (Tables, Images) of sufficient quality for clarity?

Reviewer #1: (No Response)

Reviewer #2: (No Response)

**Conclusions**

-Are the conclusions supported by the data presented?

-Are the limitations of analysis clearly described?

-Do the authors discuss how these data can be helpful to advance our understanding of the topic under study?

-Is public health relevance addressed?

Reviewer #1: (No Response)

Reviewer #2: (No Response)

**Editorial and Data Presentation Modifications?**

Reviewer #1: (No Response)

Reviewer #2: (No Response)

**Summary and General Comments**

Reviewer #1: The author answered all questions I raised. The current manuscript has allayed my concerns.

Reviewer #2: (No Response)

PLOS authors have the option to publish the peer review history of their article (what does this mean?). If published, this will include your full peer review and any attached files.

Reviewer #1: No

Reviewer #2: No

---

## [Editor Report · Acceptance letter]

26 Jul 2022

Dear Dr Liu,

We are delighted to inform you that your manuscript, "JQ-1 ameliorates schistosomiasis liver granuloma in mice by suppressing male and female reproductive systems and egg development of Schistosoma japonicum," has been formally accepted for publication in PLOS Neglected Tropical Diseases.

Best regards,

Shaden Kamhawi

co-Editor-in-Chief

Paul Brindley

co-Editor-in-Chief
